# DIFFUSION NEGATIVE PREFERENCE OPTIMIZATION MADE SIMPLE

**Joshua Tian Jin Tee**[1]     **Hee Suk Yoon**[1]     **Sunjae Yoon**[2]
**Tri Ton**[1]     **Chang D. Yoo**[1]*
[1]Korea Advanced Institute of Science and Technology (KAIST)
[2]Chung-Ang University
[1]{joshuateetj, hskyoon, tritth, cd_yoo}@kaist.ac.kr
[2]sunjaeyoon@cau.ac.kr

## ABSTRACT

Classifier-Free Guidance (CFG) improves diffusion sampling by encouraging conditional generations while discouraging unconditional ones. Existing preference alignment methods, however, focus only on positive preference pairs, limiting their ability to actively suppress undesirable outputs. Diffusion Negative Preference Optimization (Diff-NPO) approaches this limitation by introducing a separate negative model trained with inverted labels, allowing it to capture signals for suppressing undesirable generations. However, this design comes with two key drawbacks. First, maintaining two distinct models throughout training and inference substantially increases computational cost, making the approach less practical. Second, at inference time, Diff-NPO relies on weight merging between the positive and negative models, a process that dilutes the learned negative alignment and undermines its effectiveness. To overcome these issues, we introduce Diff-SNPO, a single-network framework that jointly learns from both positive and negative preferences. Our method employs a bounded preference objective to prevent winner-likelihood collapse, ensuring stable optimization. Diff-SNPO delivers strong alignment performance with significantly lower computational overhead, showing that explicit negative preference modeling can be simple, stable, and efficient within a unified diffusion framework. Code and models are available at https://github.com/JoshuaTTJ/DiffSNPO..

## 1 INTRODUCTION

Diffusion models (Ho et al., 2020) have become the backbone of modern visual content generation, achieving remarkable fidelity in synthesizing images, videos, and multimodal content (Rombach et al., 2022; Ho et al., 2022; Ruan et al., 2023; Ton et al., 2025; Hong et al., 2025a; Yoon et al., 2025b). However, models trained on vast, uncurated web-scale datasets often inherit biases and fail to align with human notions of quality, aesthetics, or safety. Consequently, fine-tuning models with human feedback through preference alignment has become a critical step for bridging the gap between a model's raw capabilities and user intent (Black et al., 2024; Wallace et al., 2024; Li et al., 2024; Tee et al., 2025).

At the heart of high-quality diffusion sampling lies *Classifier-Free Guidance (CFG)* (Ho & Salimans, 2021), a technique that enhances sample quality by amplifying the contrast between conditional and unconditional (or negatively conditioned) likelihoods. This mechanism guides generation toward prompt-aligned outputs by explicitly pushing samples away from the lower-quality unconditional distribution. However, a challenge arises with many preference optimization methods, such as Diffusion Direct Preference Optimization (Diff-DPO). These methods often apply the same optimization objective to both the conditional and unconditional branches. This uniform reinforcement of preferred attributes, while shifting the overall distribution in the desired direction, fails to heighten the critical contrast that CFG relies on. Consequently, the model's ability to generate outputs aligned with user intent remains limited.

---

*Corresponding Author

Table 1: Comparison of methods by alignment type, model setup, and use of merging strategy.

| Method | Negative Alignment | Dual Model | Merging Strategy |
|--------|:---:|:---:|:---:|
| DPO | ✗ | ✗ | ✗ |
| NPO | ✓ | ✓ | ✓ |
| CHATS | ✓ | ✓ | ✓ |
| SNPO | ✓ | ✗ | ✗ |

To further strengthen the contrastive effect of CFG, recent methods (Fu et al., 2025; Wang et al., 2025) have incorporated negative preference alignment into the training process. In particular, CHATS (Fu et al., 2025) and Diff-NPO (Wang et al., 2025) train two separate models: a "positive" model on standard preference data and a "negative" model on inverted preferences. This setup allows the system to explicitly learn from dispreferred samples and steer generation away from undesirable attributes, resulting in improved human preference alignment. Despite their effectiveness, a key limitation of both approaches lies in their reliance on distinct sets of parameter for the two models, which often leads to misaligned outputs and complicates guided sampling (Wang et al., 2025). Diff-NPO addresses this by merging the weights of the two models, whereas CHATS perturbs conditional embeddings to blend their signals. These differences are summarized in Table 1. In this work, we focus on Diff-NPO, since its sampling procedure more closely follows standard CFG. Building on this foundation, we highlight two central limitations in its design. First, training and sampling from two independent models inherently doubles both computational and memory costs, creating a significant scalability challenge. Second, merging weights that were optimized separately introduces a distribution mismatch that weakens the contribution of the negative model. This reduces the method's ability to suppress dispreferred outputs and ultimately limits the gains expected from negative alignment.

In this work, we ask: ***Can we achieve the benefits of explicit negative modeling without the costs and compromises of a dual-model architecture?*** We introduce *Diffusion Simple Negative Preference Optimization (Diffusion-SNPO)*, a framework that integrates positive and negative preference signals into a single, unified network. Our approach leverages the inherent dual-branch structure of CFG-enabled models, training the conditional branch on positive preferences and the unconditional branch on negative (inverted) preferences. This design eliminates the need for separate models and weight merging, preserving a strong, explicit contrast between preferred and dispreferred distributions within one efficient architecture.

However, we find that naively applying this strategy results in an unintended side effect: the generated images become progressively blurrier as training continues. We attribute this to the likelihood decrease observed in the DPO algorithm (Rafailov et al., 2024b; Pal et al., 2024; Cho et al., 2025), which, when coupled with conflicting gradients from flipped preferences between the conditional and unconditional branches, causes instability. As a result, the model converges to a suboptimal solution, with blurring becoming more pronounced as the likelihood of winning samples decreases. To address this, we adapt Bounded DPO (Cho et al., 2025), a preference optimization method designed to increase the likelihood of winning samples during training, to Diffusion Models. This adaptation stabilizes training by preventing the loss from being dominated by low-likelihood "losing" samples, thereby boosting the likelihood of winning samples throughout preference optimization.

**In detail, our contributions can be summarized as follows:**

- We identify the challenges of dual-model negative preference optimization (NPO), namely its high computational cost and the performance trade-offs inherent in its two separate model design.

- We demonstrate that simply applying opposing preferences on different branches within a single model leads to poor interactions with DPO-based algorithms, resulting in blurry images as training progresses.

- We introduce Diff-SNPO, a single-model negative preference optimization method for Diffusion Models. By adapting the BDPO algorithm and deriving an upper bound, our approach effectively eliminates the blurring effect caused by DPO in a negative preference optimization setting.

- We conduct extensive evaluations on the Pick-a-Pic v2 benchmark with both SD1.5 and SDXL, showing that **Diff-SNPO** delivers strong performance across multiple alignment metrics, while being more efficient in both training and inference.

## 2 RELATED WORK

### 2.1 PREFERENCE OPTIMIZATION

Preference optimization aims to align generative models with human judgments. One common strategy is to train a *reward model* that scores prompt–image pairs based on semantic or aesthetic quality, and then fine-tune the diffusion model to maximize these scores (Xu et al., 2023; Clark et al., 2024). Another direction draws on *policy optimization*, which frames denoising as a sequential decision process: DDPO (Black et al., 2024) applies reinforcement learning across the sampling steps, while DPOK (Fan et al., 2023) introduces a KL-regularized reward objective. More recently, methods that bypass explicit reward modeling by learning directly from curated positives or pairwise preferences have attracted growing interest (Wallace et al., 2024; Lu et al., 2025; Hong et al., 2025b; Tee et al., 2025; Yoon et al., 2025a). While these approaches simplify training, they typically handle negative feedback only implicitly, by training models to favor preferred outputs in relative comparisons. This indirect treatment leaves little control over explicitly suppressing undesirable generations. In contrast, explicitly modeling dispreferred outcomes offers a more direct form of control: it steers generation away from undesirable regions of the distribution, reducing artifacts and improving alignment.

### 2.2 NEGATIVE PREFERENCE OPTIMIZATION

In response to the limitations of standard preference optimization algorithms, recent work has explored dual-model strategies that explicitly separate positive and negative preferences. Diff-NPO (Wang et al., 2025) trains a negative model on inverted preferences and substitutes it for the unconditional branch during inference, which improves alignment but introduces a mismatch between training and inference due to weight interpolation. CHATS (Fu et al., 2025) addresses this issue by jointly training positive and negative models within a contrastive objective, achieving stronger alignment but at the cost of increased computation and memory from maintaining two networks. These challenges highlight the need for approaches that unify positive and negative preference modeling within a single framework. Our method, DIFF-SNPO, takes this direction by integrating both signals into a shared training objective, reducing overhead while better balancing alignment across positive and negative preferences.

## 3 PRELIMINARIES

### 3.1 DIFFUSION MODELS

Denoising Diffusion Probabilistic Models (DDPMs) (Ho et al., 2020) consist of a forward noising process and a learned reverse denoising process. The forward process gradually perturbs clean data $x_0$ with Gaussian noise according to a variance schedule $\{\beta_t\}_{t=1}^{T}$:

$$q(x_{1:T} \mid x_0) = \prod_{t=1}^{T} q(x_t \mid x_{t-1}), \quad q(x_t \mid x_{t-1}) = \mathcal{N}\left(x_t; \sqrt{1-\beta_t}\, x_{t-1}, \beta_t I\right). \tag{1}$$

To invert this process, a neural network $\epsilon_\theta$ parameterizes the reverse transitions by predicting the injected noise:

$$p_\theta(x_{t-1} \mid x_t) = \mathcal{N}\left(x_{t-1}; \tfrac{1}{\sqrt{\alpha_t}}\left(x_t - \tfrac{\beta_t}{\sqrt{1-\bar{\alpha}_t}}\, \epsilon_\theta(x_t, t)\right), \sigma_t^2 I\right), \tag{2}$$

where $\alpha_t = 1 - \beta_t$, $\bar{\alpha}_t = \prod_{s=1}^{t} \alpha_s$, and $\sigma_t^2$ denotes the variance.

Model training minimizes a variational bound, which simplifies to a weighted noise-prediction loss:

$$\mathcal{L}_{\text{DDPM}} = \mathbb{E}_{x_0, \epsilon, t}\left[\omega(t) \left\| \epsilon - \epsilon_\theta(x_t, t) \right\|^2\right], \tag{3}$$

with $\epsilon \sim \mathcal{N}(0, I)$ and $t \sim \mathcal{U}\{1, \ldots, T\}$. The weighting function $\omega(t)$ controls the relative contribution of different timesteps, reflecting the varying difficulty of denoising across the diffusion trajectory. This objective provides a simple and stable training criterion that underpins most modern diffusion-based generative models.

## 3.2 CLASSIFIER-FREE GUIDANCE

Classifier-Free Guidance (CFG) (Ho & Salimans, 2021) modifies the conditional sampling distribution to strengthen (or weaken) the influence of conditioning information $c$ while remaining anchored to the unconditional model. At timestep $t$, the effective distribution is defined as

$$\tilde{p}_\theta(x_t \mid c) \propto p_\theta(x_t)\left(\frac{p_\theta(x_t \mid c)}{p_\theta(x_t)}\right)^\omega, \qquad \omega > 0, \tag{4}$$

where $\omega = 1$ recovers the standard conditional model, $\omega > 1$ amplifies the effect of the condition, and $0 < \omega < 1$ dampens it.

In diffusion sampling, the model evolves states using the score function $s_\theta(x_t, t, c) = \nabla_{x_t} \log p_\theta(x_t \mid c)$. Relating equation 4 to the score yields

$$\tilde{s}_\theta(x_t, t, c) = s_\theta(x_t, t) + \omega\big(s_\theta(x_t, t, c) - s_\theta(x_t, t)\big), \tag{5}$$

where $s_\theta(x_t, t)$ and $s_\theta(x_t, t, c)$ denote the unconditional and conditional scores, respectively. Thus, CFG moves along the direction that separates the conditional and unconditional scores, with $\omega$ controlling how far we step in that direction.

Under the DDPM parameterization Ho et al. (2020) $x_t = \sqrt{\bar{\alpha}_t}\, x_0 + \sqrt{1 - \bar{\alpha}_t}\, \epsilon$, the score and the noise prediction are linked by

$$s_\theta(x_t, t, \cdot) = -\frac{1}{\sqrt{1 - \bar{\alpha}_t}}\, \epsilon_\theta(x_t, t, \cdot). \tag{6}$$

Substituting equation 6 into equation 5 gives the guided noise estimator

$$\tilde{\epsilon}_\theta(x_t, t, c) = \epsilon_\theta(x_t, t) + \omega\big(\epsilon_\theta(x_t, t, c) - \epsilon_\theta(x_t, t)\big). \tag{7}$$

In essence, CFG modifies the denoiser by amplifying the difference between the conditional and unconditional predictions, with $\omega$ controlling how strongly the generation is steered toward the conditional distribution. The unconditional prediction $\epsilon_\theta(x_t, t)$ serves as a reference point, and the correction term $\epsilon_\theta(x_t, t, c) - \epsilon_\theta(x_t, t)$ shifts the generation trajectory away from regions favored by the unconditional model but inconsistent with the conditioning signal $c$. This allows CFG to suppress undesirable or generic outputs that the unconditional model might produce on its own, encouraging samples that better reflect the intended conditioning. As a result, CFG offers a principled and tunable mechanism to improve text-image alignment.

## 3.3 DIFFUSION DIRECT PREFERENCE OPTIMIZATION

Preference alignment is often formalized using the Bradley–Terry (BT) model (Kendall & Smith, 1940). Building on this idea, Direct Preference Optimization (DPO) (Rafailov et al., 2023) sidesteps the need to fit an explicit reward model by defining an implicit reward through likelihood ratios between the policy $\pi_\theta$ and a reference policy $\pi_{\text{ref}}$:

$$\mathcal{L}_{\text{DPO}}(\theta) = -\mathbb{E}_{(x_0^w, x_0^l, c) \sim \mathcal{D}}\left[\log \sigma\Big(\beta\Big(\log \frac{\pi_\theta(x_0^w \mid c)}{\pi_{\text{ref}}(x_0^w \mid c)} - \log \frac{\pi_\theta(x_0^l \mid c)}{\pi_{\text{ref}}(x_0^l \mid c)}\Big)\Big)\right]. \tag{8}$$

Here, $\beta > 0$ is a temperature parameter that controls the strength of the preference signal, and $\sigma(\cdot)$ denotes the sigmoid function. Extending this objective to diffusion models requires assigning preferences over entire trajectories $x_{0:T}$. In this context, Diff-DPO (Wallace et al., 2024) derives an upper bound on the exact Diff-DPO objective:

$$\mathcal{L}_{\text{Diff-DPO}}(\theta; y) = -\mathbb{E}_{(x_0^w, x_0^l, c) \sim \mathcal{D},\, t \sim \mathcal{U}[1, T]}\left[\log \sigma\big(yT\omega(t)\beta(\Delta_t^w(c) - \Delta_t^l(c))\big)\right], \tag{9}$$

with

$$\Delta_t^w(c) = \big\|\epsilon^w - \epsilon_\theta(x_t^w, t, c)\big\|_2^2 - \big\|\epsilon^w - \epsilon_{\text{ref}}(x_t^w, t, c)\big\|_2^2, \tag{10}$$

$$\Delta_t^l(c) = \big\|\epsilon^l - \epsilon_\theta(x_t^l, t, c)\big\|_2^2 - \big\|\epsilon^l - \epsilon_{\text{ref}}(x_t^l, t, c)\big\|_2^2, \tag{11}$$

Here, $x_t^w$ and $x_t^l$ are noisy states obtained by applying the forward diffusion process to the clean samples $x_0^w, x_0^l$ with corresponding noise terms $\epsilon^w, \epsilon^l \sim \mathcal{N}(0, I)$. The label $y \in \{+1, -1\}$ encodes the preference direction and $\omega(t)$ is a time-dependent weighting function.

### 3.4 DIFFUSION NEGATIVE PREFERENCE OPTIMIZATION

Diffusion Negative Preference Optimization (Diff-NPO) (Wang et al., 2025) extends standard preference learning by explicitly modeling undesirable behavior. It does so by training on an *inverted* preference dataset, where the roles of winners and losers are swapped. This yields a negatively aligned model, $\theta^-$, which learns to assign higher likelihood to dispreferred samples. During sampling, Diff-NPO replaces the standard unconditional branch in classifier-free guidance with this negative model, leading to the following guidance formulation:

$$\epsilon_{\text{NPO}}(x_t, c) = \epsilon_{\theta^+}(x_t, c) + \omega \left[ \epsilon_{\theta^+}(x_t, c) - \epsilon_{\theta^-}(x_t, \varnothing) \right], \tag{12}$$

where $\epsilon_{\theta^+}$ is the positively aligned model trained on standard preference data, and $\omega$ is a guidance strength hyperparameter.

In practice, however, the positive and negative models that are trained independently often exhibit poor correlation, which undermines the effectiveness of $\epsilon_{\theta^-}$ as a meaningful contrastive signal during sampling. To mitigate this, Diff-NPO applies a weight-merging procedure that combines the reference model $\theta_{\text{ref}}$, the positive model $\theta^+$, and the negative model $\theta^-$:

$$\hat{\theta}^- = \theta_{\text{ref}} + \alpha(\theta^+ - \theta_{\text{ref}}) + \beta(\theta^- - \theta_{\text{ref}}), \tag{13}$$

where $\alpha$ and $\beta$ are interpolation coefficients. In practice, the merged parameters $\hat{\theta}^-$ replace $\epsilon_{\theta^-}$ in the guidance formulation of Eq. 13, yielding improved generation quality. However, this benefit comes at the cost of higher training overhead and weakened negative alignment.

## 4 METHOD

### 4.1 LIMITATIONS OF NEGATIVE PREFERENCE OPTIMIZATION

Dual-model Negative Preference Optimization (NPO) (Wang et al., 2025) trains separate models, $\theta^+$ and $\theta^-$, for positive and negative preferences. However, maintaining two networks doubles memory, compute, and training time, severely limiting scalability for large diffusion backbones.

During inference, separate models preserve preference contrast but often degrade sample quality, particularly for $\theta^-$. Diff-NPO mitigates this via a merged parameter formulation, $\hat{\theta}^-$ (Eq. 13), interpolating between $\theta^+$, $\theta^-$, and the reference backbone $\theta_{\text{ref}}$. While merging improves generation quality (Wang et al., 2025), it heavily biases the result toward $\theta^+$, diminishing the influence of $\theta^-$ (Figure 1). This fidelity-alignment trade-off reveals that relying on separate networks and post-hoc merging ultimately undermines the preference contrast NPO is designed to enforce. To resolve this, the next section introduces a robust, scalable single-model formulation that unifies both preference signals within a shared architecture.

### 4.2 ISSUES WITH NAIVE DIFF-SNPO

Standard diffusion models inherently support both conditional and unconditional branches within a single network—a design popularized by Classifier-Free Guidance (Ho & Salimans, 2021). This built-in dual-branch structure presents a natural alternative to the dual-model Diff-NPO setup: rather than maintaining two separate networks, one can leverage the existing architecture by updating the conditional branch with preferred samples and the unconditional (or negatively conditioned) branch with dispreferred ones. This approach preserves the contrast between preference signals while avoiding the redundancy and overhead of training and managing separate models.

Specifically, let $Y \in \{+1, -1\}$ be a branch label with $\Pr(Y = +1) = 1 - p$ and $\Pr(Y = -1) = p$, where

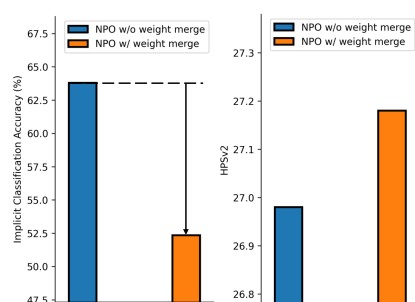

Figure 1: **Negative implicit accuracy (left) and HPSv2 (right) on SD1.5.** Weight merging lowers implicit accuracy while increasing reward, revealing a trade-off.

**Naïve Diff-SNPO**

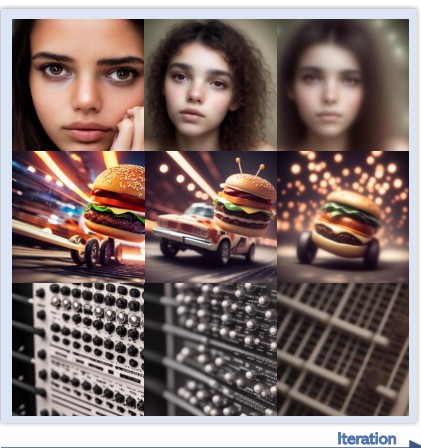

**Diff-SNPO**

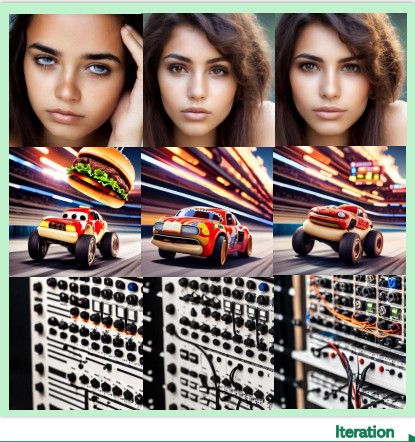

Iteration

Iteration

Figure 3: **Generated samples across training iterations.** Naive-SNPO produces progressively blurred outputs during training, while Diff-SNPO remains stable and yields increasingly preference-aligned images without blurring artifacts.

$p \in [0, 1]$ is the CFG dropout probability. The effective conditioning is

$$\tilde{c}(Y) = \begin{cases} c, & Y = +1 \quad \text{(conditional branch)} \\ \varnothing, & Y = -1 \quad \text{(unconditional/null branch)}, \end{cases}$$

where $c$ denotes the conditioning input (e.g., a text prompt) and $\varnothing$ indicates null conditioning, as in CFG. A Naive Diff-SNPO objective can thus be constructed by adapting the Diff-DPO objective in Eq. 9:

$$\mathcal{L}_{\text{Naive Diff-SNPO}}(\theta) = -\mathbb{E}_{(x_0^w, x_0^l) \sim \mathcal{D}, t, Y}\left[\log \sigma\big(Y T \omega(t) \beta \big(\Delta_t^w(\tilde{c}(Y)) - \Delta_t^l(\tilde{c}(Y))\big)\big)\right], \quad (14)$$

While Naive Diff-SNPO provides a simple and intuitive way to incorporate flipped preferences within a single network, it exhibits substantial degradation in generation quality over training. As shown in Fig. 3, this naive objective produces increasingly blurry outputs, with reduced contrast and diminished high-frequency detail. We attribute this behavior to a known property of the DPO objective: improvements in pairwise margin often come from decreasing the likelihood of both candidates—penalizing the loser more strongly—rather than consistently increasing the likelihood of the winner (Pal et al., 2024; Rafailov et al., 2024b; Cho et al., 2025). This trend is reflected in our results as well, as illustrated in Fig. 2.

To quantify this effect on preferred samples, we track the win-sample likelihood ratio against a fixed reference model, given by:

$$\frac{\pi_\theta(x^w)}{\pi_{\text{ref}}(x^w)} \approx \mathbb{E}_{x_0^w, t}\left[e^{\Delta_t^w(c)}\right].$$

Under Naive Diff-SNPO, we observe a consistent decrease in the relative win probability of preferred samples over the course of training (Fig. 3), indicating that the model suppresses likelihoods across both branches rather than reinforcing the preferred ones. This undesirable trend, combined with the structure of the optimization signals, leads to a characteristic blurring effect. Specifically, Naive Diff-SNPO ap-

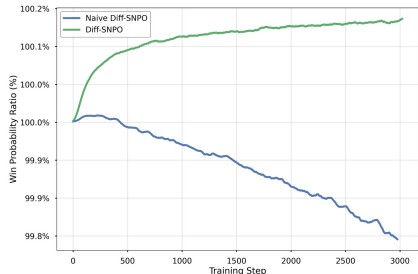

Figure 2: **Win probability ratio over training.** Naive-SNPO's win probability decreases during training, whereas Diff-SNPO's improves steadily.

plies symmetric but opposing updates to the conditional and unconditional branches—encouraging one to increase the likelihood of a sample while the other decreases it. Since both branches share model parameters, these conflicting gradients interfere with each other, and the model resolves this

conflict by averaging them. In generative settings, this averaging dampens contrast and degrades fine detail, resulting in blurry outputs. To overcome this limitation, we seek to break the destructive symmetry in the update rule. In the next section, we introduce an asymmetric preference optimization approach that biases learning toward increasing win probability, thereby mitigating gradient interference and addressing the blurring effect.

### 4.3 DIFF-SNPO

Recently, Cho et al. (2025) introduced Bounded DPO (BDPO) to address a key shortcoming of standard DPO. As the model reduces the probability assigned to the losing sample, the corresponding lose sample log-likelihood term, $\log \pi_\theta(\mathbf{y}_l \mid \mathbf{x})$, in the objective grows disproportionately large, causing the loss to become dominated by the loser. This skews the gradient signal and can even drive updates that decrease the likelihood of the preferred (winning) output, despite the preference label.

To mitigate this issue, BDPO replaces the losing term with a mixture distribution that includes a non-vanishing contribution from the reference policy:

$$\pi_{\mathrm{mix}}(\mathbf{y} \mid \mathbf{x}) = \lambda \, \pi_\theta(\mathbf{y} \mid \mathbf{x}) + (1 - \lambda) \, \pi_{\mathrm{ref}}(\mathbf{y} \mid \mathbf{x}), \qquad \lambda \in (0, 1),$$

which leads to the modified objective

$$\mathcal{L}_{\mathrm{BDPO}}(\pi_\theta; \pi_{\mathrm{ref}}) = - \mathbb{E}_{(x_0^w, x_0^l, c) \sim \mathcal{D}} \left[ \log \sigma \Big( \beta \big[ \log \tfrac{\pi_\theta(x_0^w | c)}{\pi_{\mathrm{ref}}(x_0^w | c)} - \log \tfrac{\pi_{\mathrm{mix}}(x_0^l | c)}{\pi_{\mathrm{ref}}(x_0^l | c)} \big] \Big) \right]. \tag{15}$$

This modification bounds the contribution of the loser term and prevents it from overwhelming the loss, thereby preserving the intended effect of preference optimization: promoting the winning sample. In addition, BDPO retains the same global minimizers as DPO while enforcing a lower bound on the winning likelihood, offering stronger stability guarantees throughout training (Cho et al., 2025).

To adapt BDPO to diffusion models, similar to Diff-DPO, we define the trajectory-level reward

$$r_\theta(c, \mathbf{x}_0) = \mathbb{E}_{\mathbf{x}_{1:T} \sim p_\theta(\cdot | \mathbf{x}_0, c)} \Big[ R(c, \mathbf{x}_{0:T}) \Big]. \tag{16}$$

The corresponding Diffusion BDPO objective is

$$\mathcal{L}_{\mathrm{Diff\text{-}BDPO}}(\theta) = - \mathbb{E}_{(\mathbf{x}_{0:T}^w, \mathbf{x}_{0:T}^l, c) \sim \mathcal{D}} \left[ \log \sigma \Big( \beta \big[ \log \tfrac{p_\theta(\mathbf{x}_{0:T}^w | c)}{p_{\mathrm{ref}}(\mathbf{x}_{0:T}^w | c)} - \log \tfrac{\pi_{\mathrm{mix}}(\mathbf{x}_{0:T}^l | c)}{p_{\mathrm{ref}}(\mathbf{x}_{0:T}^l | c)} \big] \Big) \right]. \tag{17}$$

Following Diff-DPO, we upper-bound this objective via its ELBO and then apply Jensen's inequality to obtain the result below (see Appendix A.2 for the full derivation):

$$\mathcal{L}_{\mathrm{Diff\text{-}BDPO\text{-}UB}}(\theta) = - \mathbb{E}_{(x_0^w, x_0^l, c) \sim \mathcal{D}, \, t \sim \mathcal{U}[1, T]} \Big[ \log \sigma \Big( - \beta \, (m(x_t^w, c) - m_{\mathrm{mix}}(x_t^l, c)) \Big) \Big]. \tag{18}$$

Here the per-step terms are:

$$d_\theta(x_t, \epsilon, t, c) = T \, \omega(t) \, \| \epsilon - \epsilon_\theta(x_t, t, c) \|_2^2, \quad d_{\mathrm{ref}}(x_t, \epsilon, t, c) = T \, \omega(t) \, \| \epsilon - \epsilon_{\mathrm{ref}}(x_t, t, c) \|_2^2, \tag{19}$$

$$m(x_t, c) = d_\theta(x_t, \epsilon, t, c) - d_{\mathrm{ref}}(x_t, \epsilon, t, c), \tag{20}$$

$$m_{\mathrm{mix}}(x_t, c) = - \log \Big( \lambda \, e^{-d_\theta(x_t, \epsilon, t, c)} + (1 - \lambda) \, e^{-d_{\mathrm{ref}}(x_t, \epsilon, t, c)} \Big) \; - \; d_{\mathrm{ref}}(x_t, \epsilon, t, c). \tag{21}$$

where $\mathcal{L}_{\mathrm{Diff\text{-}BDPO\text{-}UB}}$ denotes the upper bound approximation of Diff-BDPO.

Building on this, we define our final Diff-SNPO objective, which applies our introduced Diff-BDPO-UB objective into a single model negative preference optimization framework:

$$\mathcal{L}_{\mathrm{SNPO}}(\theta) = - \mathbb{E}_{(\mathbf{x}^w, \mathbf{x}^l, c) \sim \mathcal{D}, \, t \sim p(t), \, Y} \Big[ \log \sigma \Big( \beta \, (m(\tilde{x}^w(Y), \tilde{c}(Y)) - m_{\mathrm{mix}}(\tilde{x}^l(Y), \tilde{c}(Y))) \Big) \Big]. \tag{22}$$

where $\tilde{x}^{w/l}(Y)$ is given as:

$$\tilde{x}^{w/l}(Y) = \begin{cases} x^{w/l}, & \text{if } Y = +1 \quad \text{(conditional branch)} \\ x^{l/w}, & \text{if } Y = -1 \quad \text{(unconditional/null branch)}. \end{cases} \tag{23}$$

As shown in Fig. 3, Diff-SNPO avoids the decline in winner likelihood observed with Naive Diff-SNPO: the estimated log-likelihood of winning samples steadily improves throughout training. In line with this, Fig. 3 further shows that Diff-SNPO prevents the progressive low-contrast "blurring" artifact characteristic of the naive training objective, effectively addressing its key shortcoming.

Table 2: **Comparison of Diff-SNPO with baseline methods on SD1.5 and SDXL backbones using Pick-a-Pic v2.** All values are reported as mean $\pm$ 95% confidence interval over 4 random seeds. Diff-SNPO consistently achieves the highest scores across most human preference metrics, reflecting improved alignment and visual quality. For clarity, the best-performing method in each metric is shown in **bold**, and the second-best is underlined.

| Model | Method | HPSv2 | Pick Score | Aesthetic Score | Image Reward |
|-------|--------|-------|-----------|-----------------|--------------|
| SD1.5 | Baseline | $26.24 \pm 0.14$ | $20.64 \pm 0.09$ | $5.2849 \pm 0.12$ | $0.1221 \pm 0.08$ |
| | Diff-DPO (Wallace et al., 2024) | $26.55 \pm 0.04$ | $21.01 \pm 0.04$ | $5.3823 \pm 0.01$ | $0.2968 \pm 0.07$ |
| | Diff-NPO (Wang et al., 2025) | $26.92 \pm 0.05$ | $\underline{21.46 \pm 0.04}$ | $5.5381 \pm 0.04$ | $\underline{0.3786 \pm 0.09}$ |
| | CHATS (Fu et al., 2025) | $\underline{27.20 \pm 0.16}$ | $21.05 \pm 0.06$ | $\mathbf{5.6845 \pm 0.07}$ | $0.2995 \pm 0.07$ |
| | Diff-BDPO (Ours) | $26.64 \pm 0.16$ | $21.15 \pm 0.10$ | $5.4457 \pm 0.10$ | $0.3165 \pm 0.11$ |
| | Diff-SNPO (Ours) | $\mathbf{27.23 \pm 0.07}$ | $\mathbf{22.24 \pm 0.03}$ | $\underline{5.6258 \pm 0.04}$ | $\mathbf{0.6936 \pm 0.07}$ |
| SDXL | Baseline | $27.43 \pm 0.07$ | $22.13 \pm 0.06$ | $5.8850 \pm 0.01$ | $0.7605 \pm 0.05$ |
| | Diff-DPO (Wallace et al., 2024) | $28.09 \pm 0.03$ | $22.59 \pm 0.02$ | $\underline{5.8884 \pm 0.01}$ | $0.9841 \pm 0.07$ |
| | Diff-NPO (Wang et al., 2025) | $\underline{28.30 \pm 0.08}$ | $\underline{22.67 \pm 0.05}$ | $\mathbf{5.9449 \pm 0.02}$ | $0.9847 \pm 0.01$ |
| | CHATS (Fu et al., 2025) | $28.25 \pm 0.10$ | $22.34 \pm 0.08$ | $5.8785 \pm 0.03$ | $\mathbf{1.0543 \pm 0.03}$ |
| | Diff-BDPO (Ours) | $28.13 \pm 0.04$ | $22.36 \pm 0.10$ | $5.8037 \pm 0.07$ | $0.9946 \pm 0.05$ |
| | Diff-SNPO (Ours) | $\mathbf{28.33 \pm 0.08}$ | $\mathbf{22.69 \pm 0.06}$ | $5.8129 \pm 0.06$ | $\underline{1.0100 \pm 0.06}$ |

## 5 EXPERIMENTS

### 5.1 EXPERIMENTAL SETUP

**Datasets and Models.** We fine-tune both Stable Diffusion v1.5 (SD1.5) (Rombach et al., 2022) and Stable Diffusion XL (SDXL) (Podell et al., 2024) using our proposed Diffusion SNPO objective in Eq. 22. For training, we utilize the Pick-a-Pic v2 (Kirstain et al., 2023) corpus, a large-scale human preference dataset comprising 851,293 image pairs across 58,960 unique prompts.

**Training Details.** Models are initialized from the publicly available SD1.5 (Rombach et al., 2022) (CreativeML Open RAIL-M license) and SDXL (Podell et al., 2024) (MIT license) checkpoints. Training is conducted using the AdamW (Loshchilov & Hutter, 2019) optimizer with a learning rate of $2.048 \times 10^{-8}$. We use a batch size of 512 and train for 3,000 steps on SD1.5, and a larger batch size of 2,048 for 625 steps on SDXL. All experiments are run on 8×NVIDIA A6000 GPUs using distributed data parallelism. Total training time is approximately 12 hours for SD1.5 and 68 hours for SDXL. The regularization coefficient is set to $\beta = 2000$ for SD1.5 and $\beta = 5000$ for SDXL. Unless stated otherwise, Diff-SNPO is trained with an interpolation parameter of $\lambda = 0.9$.

**Baselines.** We compare Diff-SNPO against both negative preference optimization methods and standard alignment baselines. Specifically, we evaluate Diff-NPO (Wang et al., 2025), CHATS (Fu et al., 2025), and Diff-DPO (Wallace et al., 2024), along with the original pretrained models for SD1.5 (Rombach et al., 2022) and SDXL (Podell et al., 2024).

**Evaluation Protocol.** All models are evaluated using the DDIM (Song et al., 2021) sampler with 50 inference steps and a classifier-free guidance scale of 7.5. For Diff-NPO and CHATS, we adopt the hyperparameters specified in their released code, as these methods modify the sampling pipeline. Performance is assessed using five widely adopted reward models: PickScore (Kirstain et al., 2023), HPSv2 (Wu et al., 2023), ImageReward (Xu et al., 2023), and Aesthetics Score (Schuhmann, 2023). To account for stochasticity, we report average results across four random seeds, with each model generating 2,000 images using prompts from the Pick-a-Pic v2 test set.

### 5.2 QUANTITATIVE RESULTS

Table 2 presents quantitative comparisons of Diff-SNPO against established baselines on SD1.5 and SDXL backbones.

On SD1.5, Diff-SNPO consistently outperforms prior methods across most human preference metrics. Notably, it achieves 27.23 on HPSv2 and 22.24 on PickScore, marking a substantial improvement over both Diff-DPO and the more stable Diff-BDPO. This indicates that the gains stem not just

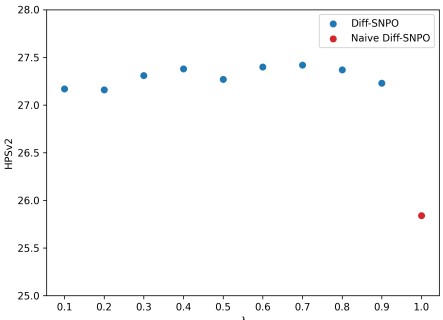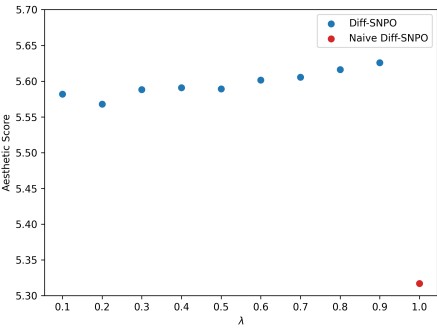

Figure 4: Ablation over $\lambda$ on SD 1.5 for both HPSv2 and Aesthetic score.

from BDPO's stabilization, but from negative preference optimization itself. While CHATS attains a slightly higher Aesthetic Score, it lags behind on all other metrics, suggesting that its more visually pleasing outputs come at the cost of semantic alignment. Diff-SNPO also surpasses its dual-model counterpart, Diff-NPO, across all metrics while using only half the computation. We attribute this to its single-model design, which preserves negative preference alignment better than Diff-NPO, where accuracy drops notably after model merging.

On SDXL, Diff-SNPO delivers strong and competitive results, with HPSv2 and ImageReward scores of 28.33 and 1.01, comparable to state-of-the-art methods. Its Aesthetic Score (5.81) is slightly below DPO (5.89), and the large advantage it shows over Diff-NPO on SD1.5 becomes smaller on SDXL. This stems from Diff-SNPO's bias toward optimizing "preferred" samples in the dataset. Although these samples are favored over their losing counterparts, they have been shown to be less aesthetically pleasing than the already strong outputs of the base SDXL model. As a result, focusing on them may slightly reduce aesthetic quality and limit gains on stronger backbones. Similar trends appear with CHATS and Diff-BDPO, suggesting the limitation comes from the dataset rather than the alignment method itself. Even so, Diff-SNPO retains important practical benefits: it requires only half the computation of Diff-NPO and, with its single-model architecture, also enables faster sampling—making it both efficient and scalable.

## 5.3 IMPACT OF THE MIXING PARAMETERS $\lambda$ AND $\beta$ ON DIFF-SNPO

In this section, we examine the impact of the parameters $\lambda$ and $\beta$ on Diff-SNPO using SD 1.5, while keeping all other hyperparameters identical to those in the main experiments.

| $\beta$ | HPSv2 | Aesthetic Score |
|---|---|---|
| 1000 | 27.30 | 5.6682 |
| 2000 | 27.23 | 5.6258 |
| 3000 | 27.23 | 5.5886 |

Table 3: Effect of $\beta$ on the Pick-a-Pic v2 dataset.

From Fig. 4, we observe that Diff-SNPO remains stable as $\lambda$ varies, showing only minor fluctuations across both reward metrics. This indicates that, within the tested range, the choice of $\lambda$ has limited impact on performance. In contrast, the setting $\lambda = 1.0$, corresponding to Naive-SNPO, shows a pronounced drop in both reward metrics. As discussed in Section 4.2, this decline stems from the blurring artifacts that emerge when the win likelihood decreases under single-model Negative Preference Optimization, ultimately degrading image quality.

When varying $\beta$, Table 3 shows that performance is similarly stable across different choices. Therefore, we retain the $\beta$ values from the original Diff-DPO configuration, as they provide a reliable default without introducing meaningful variability in performance.

## 5.4 COMPARING NEGATIVE PREFERENCE ALIGNMENT ACROSS METHODS

From Table 5, we observe that Diff-DPO and Diff-BDPO reach similar negative implicit accuracy in their unconditional branches, both consistently below 50%. This behavior is expected because their unconditional branches are trained only with positive preference alignment, which limits their ability to learn negative preference signals. In contrast, Diff-NPO experiences a substantial drop in negative implicit accuracy after weight merging, decreasing by more than 10%. Its post-merge accuracy also falls below that of Diff-SNPO and lies several points behind its own pre-merged value.

Table 4: **Training cost comparison.** Experiments were conducted on $8\times$A6000 GPUs with a total batch size of 512. Dual-model approaches require substantially more memory and incur slower training throughput. The best result in each column is shown in **bold**.

| Method | Memory (GB) $\downarrow$ | Time / Step (s) $\downarrow$ | Relative Speed $\uparrow$ |
|---|---|---|---|
| Diff-NPO | $44.2 \times 2$ | $12.25 \times 2$ | $1.00\times$ |
| CHATS | 46.3 | 13.19 | $1.86\times$ |
| **Diff-SNPO (Ours)** | **44.2** | **12.25** | **2.00**$\times$ |

Table 5: **Negative preference implicit classification accuracy and loss.** Parentheses denote Diff-NPO without weight merging. Diff-SNPO achieves higher negative implicit accuracy and lower negative preference loss than Diff-NPO, improving its negative alignment.

| Method | Neg. Implicit Acc. (%) | Neg. Pref. Loss |
|---|---|---|
| Diff-DPO | 31.86 | 0.759 |
| Diff-BDPO | 32.04 | 0.805 |
| Diff-NPO | 52.34 (63.80) | 0.703 (0.648) |
| Diff-SNPO | **57.45** | 0.668 |

Although implicit accuracy is not a definitive measure of overall performance, and can be influenced by reward-model artifacts (Rafailov et al., 2024a; Amini et al., 2024), it is still useful for revealing the sampling and training mismatch present in Diff-NPO. This mismatch is further reflected in the negative preference alignment loss, which rises sharply after weight merging. Diff-SNPO does not suffer from this problem: its single-model design preserves the negative alignment signal and avoids the degradation observed in Diff-NPO. As a result, Diff-SNPO maintains more reliable negative preference modeling and more effectively steers generation away from undesirable samples.

## 5.5 TRAINING COMPUTATIONAL COST

Beyond alignment performance, the practicality of a preference optimization algorithm also depends on its computational efficiency. To assess this, we compare the training computational cost of dual-model approaches (CHATS and Diff-NPO) against our single-model Diff-SNPO. As reported in Table 4, the single-model design of Diff-SNPO yields substantial efficiency gains in both memory usage and training time. By eliminating the need to train two separate networks, Diff-SNPO reduces memory consumption and achieves a $2\times$ speedup in per-step training time relative to its dual-model counterpart, Diff-NPO. A similar advantage is observed over CHATS: while both methods require comparable memory, Diff-SNPO trains faster because it optimizes the conditional and unconditional branches in parallel, whereas CHATS processes them sequentially. In summary, Diff-SNPO combines lower memory overhead with faster training, establishing it as a more efficient and scalable alternative for negative preference optimization. *A detailed comparison of inference cost, which further highlights the efficiency of our single-model approach, can be found in Appendix A.7.*

## 6 CONCLUSION

In conclusion, we propose Diff-SNPO, a single-model framework for Negative Preference Optimization that achieves strong performance while simplifying the training pipeline. In contrast to prior approaches that require two separate models, Diff-SNPO unifies conditional and unconditional branches within a single architecture, thereby eliminating redundant computation and substantially improving efficiency in both training and inference. Our experiments demonstrate that this streamlined design not only matches, but often exceeds the performance of existing methods, highlighting its effectiveness in preserving negative preference alignment. Beyond raw performance, the reduced computational footprint and faster sampling make Diff-SNPO a practical and scalable solution, lowering the barriers to applying preference optimization in real-world generative modeling tasks.

## 7  ACKNOWLEDGEMENTS

This work was supported by Institute for Information & communications Technology Planning & Evaluation (IITP) grant funded by the Korea government (MSIT) (No. RS-2021-II211381, Development of Causal AI through Video Understanding and Reinforcement Learning, and Its Applications to Real Environments) and partly supported by Institute of Information & communications Technology Planning & Evaluation (IITP) grant funded by the Korea government (MSIT) (No. RS-2022-II220184, Development and Study of AI Technologies to Inexpensively Conform to Evolving Policy on Ethics).

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

# A APPENDIX

## A.1 THE USE OF LLMS

We used LLMs solely for light editing such as correcting grammatical errors and polishing some words. They did not contribute to research ideation, experiments, analysis, or substantive writing.

## A.2 DERIVATION OF THE DIFFUSION-BDPO UPPER BOUND

We begin by recalling the Diffusion-BDPO objective:

$$\mathcal{L}_{\text{Diff-BDPO}}(\theta) = -\mathbb{E}_{(\mathbf{x}_{0:T}^w, \mathbf{x}_{0:T}^l, c) \sim \mathcal{D}} \left[ \log \sigma \Big( \beta \Big[ \log \frac{p_\theta(\mathbf{x}_{0:T}^w \mid c)}{p_{\text{ref}}(\mathbf{x}_{0:T}^w \mid c)} - \log \frac{p_{\text{mix}}(\mathbf{x}_{0:T}^l \mid c)}{p_{\text{ref}}(\mathbf{x}_{0:T}^l \mid c)} \Big] \Big) \right], \quad (24)$$

where the mixture distribution is defined as

$$p_{\text{mix}}\big(\mathbf{x}_{0:T}^l \mid c\big) = \lambda\, p_\theta\big(\mathbf{x}_{0:T}^l \mid c\big) + (1 - \lambda)\, p_{\text{ref}}\big(\mathbf{x}_{0:T}^l \mid c\big), \qquad \lambda \in (0, 1). \quad (25)$$

To obtain a computationally tractable form, we first decompose $\log p_\theta(\mathbf{x}_{0:T} \mid c)$ using the backward process of diffusion models:

$$\log p_\theta(\mathbf{x}_{0:T} \mid c) = \log p_\theta(\mathbf{x}_T \mid c) + \log \prod_{t=1}^{T} p_\theta(\mathbf{x}_{t-1} \mid \mathbf{x}_t, c) \quad (26)$$

$$= \log p_\theta(\mathbf{x}_T \mid c) + \sum_{t=1}^{T} \log p_\theta(\mathbf{x}_{t-1} \mid \mathbf{x}_t, c) \quad (27)$$

$$= C - \sum_{t=1}^{T} \omega(t) \left\| \epsilon_\theta(x_t, c, t) - \epsilon \right\|_2^2, \quad (28)$$

where $C$ is a parameter-independent constant.

Next, consider the mixed log-probability:

$$\log p_{\text{mix}}\big(\mathbf{x}_{0:T}^l \mid c\big) = \log\Big( \lambda\, p_\theta\big(\mathbf{x}_{0:T}^l \mid c\big) + (1 - \lambda)\, p_{\text{ref}}\big(\mathbf{x}_{0:T}^l \mid c\big) \Big)$$

$$= C + \log\Big( \lambda\, e^{\sum_{t=1}^{T} \log p_\theta(\mathbf{x}_{t-1}^l \mid \mathbf{x}_t, c)} + (1 - \lambda)\, e^{\sum_{t=1}^{T} \log p_{\text{ref}}(\mathbf{x}_{t-1}^l \mid \mathbf{x}_t, c)} \Big). \quad (29)$$

For brevity, define

$$a_t := \log p_\theta(\mathbf{x}_{t-1}^l \mid \mathbf{x}_t, c), \qquad b_t := \log p_{\text{ref}}(\mathbf{x}_{t-1}^l \mid \mathbf{x}_t, c), \qquad t = 1, \ldots, T. \quad (30)$$

Then equation 29 becomes

$$\log p_{\text{mix}}\big(\mathbf{x}_{0:T}^l \mid c\big) = C + \log\Big( \lambda\, e^{\sum_{t=1}^{T} a_t} + (1 - \lambda)\, e^{\sum_{t=1}^{T} b_t} \Big). \quad (31)$$

Applying the discrete $n$-factor Hölder inequality yields

$$\lambda e^{\sum_{t=1}^{T} a_t} + (1 - \lambda) e^{\sum_{t=1}^{T} b_t} \leq \prod_{t=1}^{T} \Big( \lambda e^{p_t a_t} + (1 - \lambda) e^{p_t b_t} \Big)^{1/p_t}, \quad (32)$$

for exponents $p_1, \ldots, p_T \geq 1$ with $\sum_{t=1}^{T} 1/p_t = 1$.

Taking logs and substituting back gives

$$\log p_{\text{mix}}\big(\mathbf{x}_{0:T}^l \mid c\big) \leq C + \sum_{t=1}^{T} \frac{1}{p_t} \log\Big( \lambda e^{p_t a_t} + (1 - \lambda) e^{p_t b_t} \Big). \quad (33)$$

Choosing $p_t = T$ for all $t$ (so that $\sum_t 1/p_t = 1$) produces the uniform bound

$$\log p_{\text{mix}}\big(\mathbf{x}_{0:T}^l \mid c\big) \leq C + \frac{1}{T} \sum_{t=1}^{T} \log\Big( \lambda e^{T \log p_\theta(\mathbf{x}_{t-1}^l \mid \mathbf{x}_t, c)} + (1 - \lambda)\, e^{T \log p_{\text{ref}}(\mathbf{x}_{t-1}^l \mid \mathbf{x}_t, c)} \Big). \quad (34)$$

Table 6: **Comparison of Diff-SNPO with baseline methods on SD1.5 and SDXL backbones on HPDv2.** All values are reported as mean $\pm$ 95% confidence interval over 4 random seeds. Diff-SNPO achieves the best scores across most human preference metrics, indicating superior alignment and visual quality. For each metric, the top-performing method is **bolded**, while the second-best is underlined.

| Model | Method | HPSv2 | Pick Score | Aesthetic Score | Image Reward |
|-------|--------|-------|------------|-----------------|--------------|
| SD1.5 | Baseline | $26.52 \pm 0.10$ | $20.80 \pm 0.06$ | $5.3406 \pm 0.17$ | $-0.0521 \pm 0.19$ |
| | Diff-DPO (Wallace et al., 2024) | $26.85 \pm 0.03$ | $21.26 \pm 0.02$ | $5.4716 \pm 0.14$ | $0.1620 \pm 0.08$ |
| | Diff-NPO (Wang et al., 2025) | $27.18 \pm 0.06$ | $\underline{21.71 \pm 0.05}$ | $5.5842 \pm 0.07$ | $0.2287 \pm 0.14$ |
| | CHATS (Fu et al., 2025) | $\underline{27.68 \pm 0.07}$ | $21.45 \pm 0.04$ | $\mathbf{5.8605 \pm 0.07}$ | $0.3699 \pm 0.08$ |
| | Diff-BDPO (Ours) | $27.24 \pm 0.03$ | $21.57 \pm 0.03$ | $5.5677 \pm 0.06$ | $\underline{0.3576 \pm 0.02}$ |
| | Diff-SNPO (Ours) | $\mathbf{27.85 \pm 0.06}$ | $\mathbf{22.86 \pm 0.03}$ | $\underline{5.7737 \pm 0.06}$ | $\mathbf{0.8093 \pm 0.08}$ |
| SDXL | Baseline | $28.01 \pm 0.15$ | $22.80 \pm 0.09$ | $5.9887 \pm 0.08$ | $0.8781 \pm 0.11$ |
| | Diff-DPO (Wallace et al., 2024) | $28.55 \pm 0.09$ | $23.12 \pm 0.01$ | $6.0296 \pm 0.05$ | $1.0788 \pm 0.09$ |
| | Diff-NPO (Wang et al., 2025) | $\underline{28.87 \pm 0.03}$ | $\underline{23.29 \pm 0.05}$ | $\mathbf{6.0816 \pm 0.01}$ | $1.1305 \pm 0.04$ |
| | CHATS (Fu et al., 2025) | $28.82 \pm 0.07$ | $22.85 \pm 0.01$ | $5.9868 \pm 0.03$ | $\underline{1.1165 \pm 0.02}$ |
| | Diff-BDPO (Ours) | $28.62 \pm 0.04$ | $22.96 \pm 0.10$ | $5.9562 \pm 0.09$ | $1.0911 \pm 0.10$ |
| | Diff-SNPO (Ours) | $\mathbf{28.89 \pm 0.09}$ | $\mathbf{23.34 \pm 0.03}$ | $5.9876 \pm 0.03$ | $\mathbf{1.1460 \pm 0.04}$ |

**Upper bound on the objective.** Substituting equation 34 and equation 28 into equation 24, we obtain

$$\mathcal{L}_{\text{Diff-BDPO}}(\theta) \leq -\mathbb{E}_{(\mathbf{x}_{0:T}^w, \mathbf{x}_{0:T}^l, c) \sim \mathcal{D}} \Big[ \log \sigma \Big( \mathbb{E}_{t \sim \mathcal{U}[1,T]} \big[ -\beta\, m(x_t^w, c) - m_{\text{mix}}(x_t^l, c) \big] \Big) \Big] \quad (35)$$

$$\leq -\mathbb{E}_{(x_0^w, x_0^l, c) \sim \mathcal{D},\, t \sim \mathcal{U}[1,T]} \Big[ \log \sigma \Big( -\beta\, (m(x_t^w, c) - m_{\text{mix}}(x_t^l, c)) \Big) \Big]$$

$$= \mathcal{L}_{\text{Diff-BDPO-UB}}(\theta), \qquad \text{(by Jensen's inequality)}. \quad (36)$$

Here the per-step terms are defined as

$$d_\theta(x_t, \epsilon, t, c) = T\, \omega(t)\, \|\epsilon - \epsilon_\theta(x_t, t, c)\|_2^2, \quad d_{\text{ref}}(x_t, \epsilon, t, c) = T\, \omega(t)\, \|\epsilon - \epsilon_{\text{ref}}(x_t, t, c)\|_2^2, \quad (37)$$

$$m(x_t, c) = d_\theta(x_t, \epsilon, t, c) - d_{\text{ref}}(x_t, \epsilon, t, c), \quad (38)$$

$$m_{\text{mix}}(x_t, c) = -\log\Big( \lambda\, e^{-d_\theta(x_t, \epsilon, t, c)} + (1-\lambda)\, e^{-d_{\text{ref}}(x_t, \epsilon, t, c)} \Big) - d_{\text{ref}}(x_t, \epsilon, t, c). \quad (39)$$

In summary, by decomposing the diffusion likelihood and applying Hölder's inequality, we obtain a tractable upper bound on the original Diffusion-BDPO objective, expressed in terms of per-step denoising errors equation 37–equation 39.

## A.3 ADDITIONAL QUANTITATIVE RESULTS

Table 6 presents the results on the HPDv2 test set, showing trends consistent with those observed on the Pick-a-Pic benchmark. On the SD1.5 backbone, Diff-SNPO delivers a clear improvement, raising HPSv2 to 22.86 and ImageReward to 0.81, underscoring its effectiveness in capturing human preferences. On the more advanced SDXL backbone, it continues to perform strongly, remaining competitive with the leading approach, Diff-NPO. Crucially, Diff-SNPO achieves this performance with significantly lower computational cost and faster sampling, benefits enabled by its single-network design. This efficiency advantage highlights its practicality, offering state-of-the-art alignment quality while maintaining scalability and runtime efficiency.

## A.4 ABLATION STUDY ON DIFFERENT ODE SOLVERS

In this section, we conduct an ablation study to assess the performance of Diff-SNPO across different ODE solvers. To this end, we evaluate four widely used solvers: DDIM (Song et al., 2021), Euler Discrete (Karras et al., 2022), UniPC (Zhao et al., 2023), and DPM Solver (Lu et al., 2022). The results, presented in Table 7, show the performance of Diff-SNPO on the Pick-a-Pic v2 dataset with SD1.5. From these results, we observe that the choice of solver has little impact on the performance metrics. This outcome aligns with the theory behind Diffusion ODE solvers, where different solvers

Table 7: Performance comparison of different samplers across various reward metrics.

| Sampler | Hpsv2 ↑ | Aesthetic Score ↑ |
|---|---|---|
| DDIM | 27.23 | 5.6258 |
| Euler Discrete | 27.22 | 5.6460 |
| UniPC | 27.25 | 5.6468 |
| Dpm Solver | 27.25 | 5.6466 |

are alternative numerical methods for solving the same ODE system (Lu et al., 2022; Karras et al., 2022). Consequently, with a sufficiently large number of function evaluations (NFE), all solvers converge to the same image, leading to negligible differences in performance.

## A.5 SAFETY ALIGNMENT RESULTS

Table 8: Comparison of IP values before and after finetuning on a CoProv2 dataset.

| SafetyDPO | IP ↓ |
|---|---|
| Baseline | 0.4308 |
| Diff-DPO | 0.5109 |
| Diff-NPO | 0.4203 |
| Diff-SNPO | 0.4719 |
| **After Finetuning on CoProv2** | |
| Diff-DPO | 0.1713 |
| Diff-NPO | 0.1318 |
| Diff-SNPO | 0.1100 |

Table 8 presents the results of training and evaluating models on the CoProv2 (Wu et al., 2025) for SD 1.5. This dataset contains 23,690 pairs of safe and unsafe images, spanning across 7 categories ((Hate, Harassment, Violence, Self-Harm, Sexual, Shocking, Illegal). Evaluation was performed using the Inappropriate Probability (IP) metric (Schramowski et al., 2023), which quantifies the model's ability to generate safe content when prompted with unsafe prompts.

From Table 8, we observe that models trained on the Pick-a-Pic v2 dataset exhibit weaker safety performance. Specifically, their IP scores increase relative to the baseline after finetuning, indicating a decline in safety. This is primarily due to the nature of the Pick-a-Pic v2 dataset itself, which contains some unsafe images within its "win" samples. As a result, models trained on this dataset are inadvertently exposed to unsafe content, causing them to generate unsafe outputs.

In contrast, when explicitly trained with a safety-oriented dataset, the safety performance improves significantly. In particular, Diff-SNPO outperforms Diff-DPO, achieving a safety score of 0.11, which is slightly better than Diff-NPO's score of 0.13. This demonstrates that using a safety-focused dataset, in combination with Diff-SNPO, can lead to improved alignment with safety objectives.

## A.6 TOTAL TRAINING COMPUTE COMPARISONS

Table 9 shows the total GPU hours required by different baseline methods and Diff-SNPO for both SD1.5 and SDXL using A6000 GPUs. The GPU hours for the baselines were calculated based on the hyperparameters specified in the respective papers. From Table 9, it is clear that Diff-SNPO outperforms its counterparts in computational efficiency, demonstrating its ability to achieve strong performance with significantly lower resource requirements.

Table 9: Total GPU hours for different methods on SD 1.5 and SDXL using A6000 GPUs..

| Model | Method | Total GPU Hours |
|---|---|---|
| SD 1.5 | Diff-DPO | 206 |
| | CHATS | 237 |
| | Diff-NPO | 2×206 |
| | Diff-SNPO | 79 |
| SDXL | Diff-DPO | 1592 |
| | CHATS | 2162 |
| | Diff-NPO | 2×1592 |
| | Diff-SNPO | 498 |

## A.7 INFERENCE COMPUTATIONAL COST

Table 10 reports the inference throughput of Diff-SNPO and other negative preference optimization methods on SD1.5. The efficiency of the single-model design is particularly evident at inference: Diff-SNPO achieves the highest throughput among all methods, processing more than twice as many images per second as the dual-model baselines.

Table 10: **Inference cost comparison.** Single-image inference cost with SD1.5 on a single A6000 GPU.

| Method↓ | Throughput (img/s) ↑ |
|---|---|
| CHATS | 0.11 |
| DiffNPO | 0.27 |
| **Diff-SNPO (Ours)** | **0.48** |

By contrast, CHATS and Diff-NPO suffer from slower inference due to the need for two sequential forward passes—one for each model—combined with additional passes required by their sampling schemes. These results underscore Diff-SNPO's ability to perform fast, parallel inference without compromising alignment quality.

Overall, our findings demonstrate that Diff-SNPO effectively balances preference alignment with computational efficiency. Its single-model design reduces training cost while delivering faster inference, making it well-suited for large-scale training as well as real-time or resource-constrained deployment scenarios.

## A.8 QUALITATIVE RESULTS

Figure 5 and Figure 6 present additional image samples generated by Diff-SNPO and various preference alignment methods on SD 1.5 and SDXL using prompts from HPDv2 test set respectively.

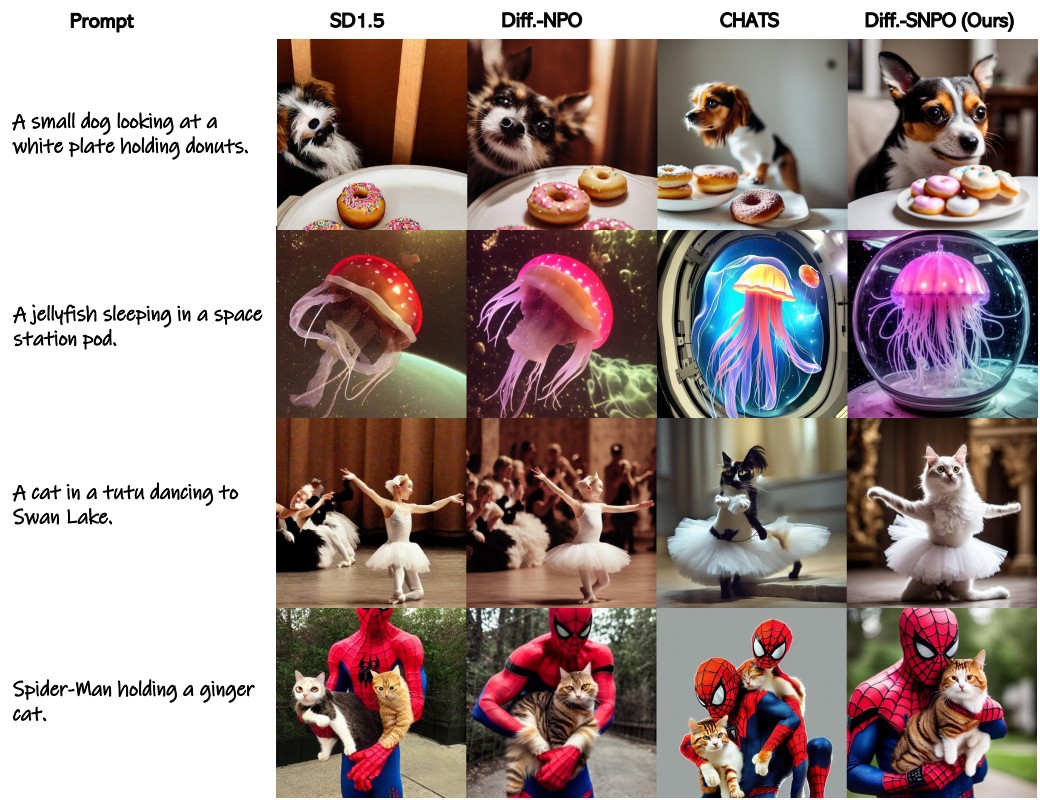

Figure 5: Side-by-side comparison of images generated by related methods on HPDv2 using SD1.5.

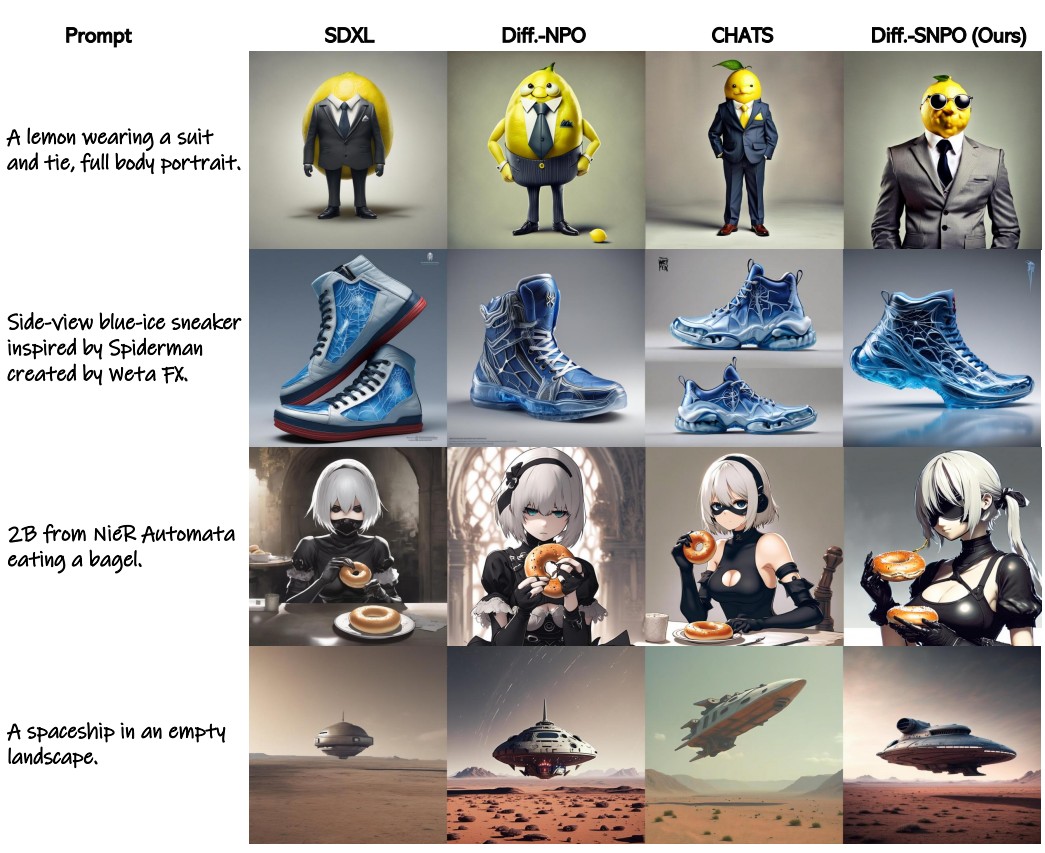

Figure 6: Side-by-side comparison of images generated by related methods on HPDv2 using SDXL.

