# OpenReview forum: "Diffusion Negative Preference Optimization Made Simple"
_ICLR.cc/2026/Conference — ICLR 2026 Poster_

### Official Review · Reviewer_ascV · 2025-10-29

**Soundness:** 2
**Presentation:** 3
**Contribution:** 2
**Rating:** 2
**Confidence:** 4

**Summary:**

This paper propose a new method called Diff-SNPO for improving preference alignment in diffusion models. It builds on Diff-NPO which uses two separate models for positive and negative preferences but that has high cost and need weight merging at inference. Instead, authors integrate both into single network using the conditional and unconditional branches from CFG. They identify problem with naive approach leading to blurry images due to likelihood collapse in DPO, and fix it by adapting Bounded DPO to diffusion setting. Experiments on SD1.5 and SDXL with Pick-a-Pic v2 show better performance and efficiency compare to baselines like Diff-NPO and CHATS.

**Strengths:**

- The idea of unifying positive and negative preference in one model is clever and adress the practical issues of dual-model approaches, like double compute and merging trade-offs. This make it more scalable for large models.

- They provide good analysis of why naive single-model fail (blurring from conflicting gradients and DPO's winner likelihood decrease), and the bounded objective seem to solve it effectively based on figures.

- Experiments are solid, evaluating on multiple reward models (PickScore, HPSv2 etc.) and showing gains in alignment metrics while reduce overhead. Code will be released which is plus.

- The derivation of Diff-BDPO upper bound is detailled in appendix, and overall contribute to stable preference optimization in diffusion.

**Weaknesses:**

- The paper claim strong performance but evaluation is only on Pick-a-Pic v2 benchmark, which is aesthetics-focused. Would be better to test on other datasets like safety or bias alignment to show generality, especialy since abstract mention safety.

- Hyperparameters like $\lambda=0.9$ and $\beta=2000/5000$ are tuned but not much ablation on them. How sensitive is method to these? Also, training time is given (12h for SD1.5) but no direct comparison of GPU hours vs baselines.

- While they discuss blurring in naive approach, the qualitative examples in Fig3 are limited to one prompt. More diverse visuals would strengthen the claim.

**Questions:**

- In Eq18, how do you sample Y during training? Is p fixed or annealed?

- Does Diff-SNPO work with other samplers beyond DDIM or only tested with that?

- Have you try applying this to non-image domains, like video diffusion?

---

> ### Author Response · Authors · 2025-11-24
> **Response to Reviewer ascV**
>
> Thank you for your comments. We've just uploaded a new version of the paper in which we incorporated your suggestions. Below we respond to your comments point by point:
>
> ### **Weaknesses (W)**
> **[W-Q1] Lack of Evaluation on non-Aesthetic Datasets (e.g., safety benchmarks).**
>
> **[W-A1]** Thank you for the insightful suggestion. To evaluate generality beyond aesthetics, we conducted additional experiments on the recent CoProv2 safety benchmark[1]. We found that Diff-SNPO, when trained solely on Pick-a-Pic v2, does not exhibit strong safety performance. However, we attribute this to a limitation of the Pick-a-Pic v2 dataset, which contains many unsafe images among its preferred (winning) samples. This undermines its effectiveness for learning safe generation behavior. To assess Diff-SNPO's true potential in safety alignment, we trained it explicitly for safety. Under this setting, Diff-SNPO outperforms Diff-DPO on CoProv2, highlighting its ability to generalize to safety-critical domains when given appropriate supervision. We include a detailed discussion of this result in `Appendix A.5`  of the revised manuscript.
>
> **[W-Q2] Lack of fine-grained ablation on $\lambda$ and $\beta$ hyperparameters, and missing comparison of total GPU hours with baselines**
>
> **[W-A2]** Thank you for the helpful suggestion. We have added a more detailed ablation study on the $\lambda$ and $\beta$ hyperparameters in `Section 5.3`. Our results show that Diff-SNPO is relatively robust to variations in both parameters. In practice, we use the same $\beta$ value as Diff-DPO to ensure consistency and a fair comparison. Additionally, we now report the total training time (in GPU hours) for all methods in `Appendix A.6`, enabling a direct comparison of training efficiency with baselines.
>
>
> **[W-Q3] Request for more examples of the blurring issue in `Figure 3`**
>
> **[W-A3]** Thank you for the suggestion. We have added additional qualitative examples in `Figure 3` to better illustrate the blurring artifacts observed in Naive-SNPO.
>
> ---
> ### **Questions (Q)**
>
> **[Q-Q1] In Eq. (18), how is $Y$ sampled during training? Is $p$ fixed or annealed?**
>
> **[Q-A1]** Similar to standard CFG training, we use a fixed value of $p$ throughout training. Specifically, we set $p$ to match the dropout rate used in Diff-DPO, ensuring consistency with prior work.
>
> **[Q-Q2] Does Diff-SNPO work with other samplers beyond DDIM?**
>
> **[Q-A2]** Yes, we have added results using four different samplers, DDIM, Euler Discrete, UniPC, and DPM Solver, in `Appendix A.4`. Our findings show that performance remains largely consistent across samplers. We attribute this to the fact that these samplers are all numerical solvers of the same underlying ODE, and thus tend to produce similar outputs when allocated a sufficient Number of Function Evaluations (NFE).
>
> **[Q-Q3] Have you try applying this to non-image domains, like video diffusion?**
>
> **[Q-A3]** Diff-SNPO relies on access to human preference data for effective training. At present, there are no large-scale, publicly available preference datasets for recent video diffusion models. Consequently, we have not evaluated Diff-SNPO in the video domain. We view this as a promising avenue for future work as suitable datasets become available.
>
> **[1] SafetyDPO: Scalable Safety Alignment for Text-to-Image Generation**

---

### Official Review · Reviewer_cffW · 2025-10-31

**Soundness:** 3
**Presentation:** 2
**Contribution:** 3
**Rating:** 6
**Confidence:** 3

**Summary:**

The paper proposes Diffusion Simple Negative Preference Optimization (Diff-SNPO), which trains a single diffusion model on a preference dataset using an upper bound of the Bounded Direct Preference Optimization (BDPO) objective and a modified classifier-free guidance (CFG) dropout training setup where the unconditional branch is the other example in the preference pair. The paper shows that training a single model with the Diff-DPO objective causes generated images to get blurrier over training (and positive example probabilities to go down), and that switching to their derived upper bound of the BDPO objective mitigates this failure.

The paper compares Diff-SNPO with 1. Diffusion DPO (Diff-DPO), which trains a single model on an upper bound of the DPO objective; 2. Diffusion Negative Preference Optimization (Diff-NPO), which uses the Diff-DPO objective to train two separate models, one with pairs flipped, followed by CFG during sampling using a merged weights model instead of the trained negative model directly; and 3. CHATS, which uses the Diff-DPO objective to train two separate models simultaneously, one for each reward term in the reward margin of the objective, followed by a variant of CFG using the two models and a proxy prompt for the negative model. Diff-SNPO is more efficient than #2 and #3 due to its use of a single model, and generally improves over Diff-DPO in quality.

**Strengths:**

1. The derivation of a bound of the BDPO loss, as well as the CFG-like training setup for training preference pairs on a single model, seem like clear contributions and are to my knowledge novel.
2. The proposed method has a clear memory/computational wins over the two-model methods, and the move from DPO to BDPO when training a single model is also nicely showcased.
3. The additional row showing Diff-BDPO (which if I understand correctly is meant to offer a direct comparison to Diff-DPO) is helpful as an ablation.

**Weaknesses:**

1. The quantitative results do not show report error bars or information about statistical significance. Also, there is not a clear winner across the quality metrics.
2. The paper could benefit from some revisions of the writing (e.g., in the abstract, introduction, related work, figure 1) to better explain what currently exists and distinguish its contributions. For instance, lines 78-79 make claims about Diff-NPO that I first thought were just missing references to past works but later realized are actually new findings from the paper. Moreover, the abstract and figure 1 focus on Diff-NPO when it's not clear why that prior work should be privileged over the other methods for comparison. (The paper does compare to others, but it might be easier to, for instance, have a table comparing this method to various others rather than focus on Diff-NPO in multiple places.)
3. The implicit classification results don't seem very relevant for any actual notion of performance that matters for these models. For instance, [Rafailov et al 2024](https://arxiv.org/abs/2406.02900) (as well as others) show the lack of correspondence between better rewards and better win rate. (This can be explained by that the reward looks at offline data that need mot characterize the model well.)

**Questions:**

1. Are the quantitative quality results significant? Could the authors include some form of error bars, significance tests, etc.?
2. Is there a reason why Diff-NPO is the main method being compared against in the story of the paper?
3. Why should we care about implicit accuracy at all (e.g., as a downside of weight merging)?

---

> ### Author Response · Authors · 2025-11-24
> **Response to Reviewer cffW**
>
> Thank you for your comments. We've just uploaded a new version of the paper in which we incorporated your suggestions. Below we respond to your comments point by point:
>
> ### **Weaknesses (W) and Questions (Q)**
> **[W&Q-Q1] Lack of Error bars in Quantitative results**
>
> **[W&Q-A1]** Thank you for the suggestion. We have updated our results to include error bars representing 95% confidence intervals for all reported metrics. As shown, Diff-SNPO consistently outperforms Diff-DPO, and overall achieves stronger performance than both CHATS and NPO, even when accounting for statistical variability. In addition to performance, it is worth highlighting that Diff-SNPO operates with a single model architecture, in contrast to the dual-model setups used in CHATS and NPO. This design contributes to improved training and inference efficiency, as detailed in `Section 5.5` and `Appendix A.7`.
>
>
> **[W&Q-Q1] Clarification on choice of Diff-NPO as primary baseline and need for clearer positioning of contributions.**
>
> **[W&Q-A2]** Thank you for the thoughtful feedback. While methods like CHATS incorporate negative preferences during training, they also modify the sampling pipeline, for example by perturbing the conditional embedding, thereby introducing sampling pipeline changes. In contrast, Diff-SNPO preserves the original sampling process, making it a more natural extension of Diff-NPO. This motivated our decision to frame Diff-NPO as the primary baseline throughout the paper. To address your suggestion, we have revised the introduction and added `Table 1` to more clearly articulate the distinctions between existing methods and our contributions, and to better position CHATS in the context of related work.
>
>
> **[W&Q-Q3] Why is implicit accuracy relevant (e.g., in the context of weight merging)?**
>
> **[W&Q-A3]** While implicit accuracy is not a direct measure of alignment quality due to risks such as reward hacking [1], we include it to illustrate the problem of mismatch between training and sampling behavior. To make this clearer, we also report the negative preference loss in `Table 5`. These results show a substantial increase in the negative preference loss after weight merging, indicating that the model’s behavior during sampling diverges from its training objective. We include this discussion in the updated manuscript
>
> **[1] Scaling Laws for Reward Model Overoptimization in Direct Alignment Algorithms**

---

### Official Review · Reviewer_f2zF · 2025-10-31

**Soundness:** 3
**Presentation:** 3
**Contribution:** 3
**Rating:** 6
**Confidence:** 3

**Summary:**

This paper proposes Diffusion Simple Negative Preference Optimization (Diff-SNPO), a single-network alternative to dual-model negative preference methods (e.g., Diff-NPO). The key idea is to train the conditional branch on preferred samples and the unconditional/null branch on inverted (negative) preferences within the same diffusion backbone (leveraging CFG’s two branches), thereby avoiding the cost and alignment dilution of maintaining and merging two models. The authors diagnose that a naive single-model version coupled with Diff-DPO blurs images by decreasing winner likelihood; they then adapt Bounded DPO (BDPO) to diffusion and derive an upper-bound (Diff-BDPO-UB) objective that mixes the loser term with a reference policy to prevent the winner likelihood collapse. The final objective (Eq. 18) instantiates SNPO with the BDPO upper bound and a mixing parameter λ. Experiments on Pick-a-Pic v2 and HPDv2 with SD1.5/SDXL show improved preference metrics (HPSv2, PickScore, ImageReward), higher negative implicit accuracy, and 2× training/inference speedups versus dual-model baselines.

**Strengths:**

1. I think the approach that uses the native conditional/unconditional branches to encode positive vs. negative preferences is both elegant and practical.
2. Adapting BDPO to diffusion and deriving a tractable upper bound yields a training signal that boosts the winner without exploding loser gradients.
3. Good empirical results.
4. Higher negative implicit accuracy than Diff-NPO post-merge, addressing a known weakness of merging

**Weaknesses:**

1. The main experiments use large batches and multi-GPU (e.g., 8×A6000, batch 512/2048), whereas Table 3's cost is a single-GPU batch-8 microbenchmark.
2. Training negative signals into the unconditional branch might introduce risks altering the unconditional anchor used by CFG in safety contexts.

**Questions:**

N/A

---

> ### Author Response · Authors · 2025-11-24
> **Response to Reviewer f2zF**
>
> Thank you for your comments. We've just uploaded a new version of the paper in which we incorporated your suggestions. Below we respond to your comments point by point:
>
> ### **Weaknesses (W)**
>
> **[W-Q1] Inconsistency Between Main Experimental Setup and Training Cost Reporting in `Table 4`**
>
> **[W-A1]** Thank you for pointing this out. We identified a typo in the description of `Table 4`. The reported training time was measured using a single GPU with batch size 8 and gradient accumulation of 8 (i.e., effective batch size 64), not just batch size 8 as originally stated. In practice, we train using 8×A6000 GPUs with batch size 8 per GPU and gradient accumulation of 8, giving a total effective batch size of 512. We have updated the table to include the training cost under this full configuration. Since multi-GPU communication overhead is minimal, the training time and efficiency gains reported remain consistent with the original findings.
>
> **[W-Q2] Concern about training negative signals into the unconditional branch and its impact on CFG behavior in safety-sensitive contexts.**
>
> **[W-A2]** Thank you for raising this important concern. In classifier-free guidance (CFG), sampling is guided away from the unconditional prediction. When the unconditional branch is trained to reflect undesirable generations, the guidance mechanism can more effectively steer the model away from unsafe content and toward safer outputs. To assess this, we include additional results on the CoProv2 safety benchmark[1] (see `Appendix A.5`). We observe that Diff-SNPO outperforms Diff-DPO, achieving a lower Inappropriate Probability (IP) score, which supports the claim that incorporating negative signals in this manner improves safety alignment without compromising CFG behavior.
>
> **[1] SafetyDPO: Scalable Safety Alignment for Text-to-Image Generation**

---

### Official Review · Reviewer_tGQL · 2025-11-04

**Soundness:** 3
**Presentation:** 3
**Contribution:** 2
**Rating:** 4
**Confidence:** 2

**Summary:**

This paper proposes Diff-SNPO, a unified framework for diffusion models to learn from positive and negative preference examples. Diff-SNPO improves upon Diff-NPO by only using one model, removing the potentially undesirable model merging step and making the procedure more computationally efficient. Experiments show that Diff-SNPO outperforms Diff-NPO and other baselines.

**Strengths:**

1. The paper has extensive empirical verification, and good results showing that Diff-SNPO outperforms other baselines.
2. The proposed method reduces training and inference cost by 2x, which can be significant when the models are large-scale.
3. The paper is mostly clearly written.

**Weaknesses:**

1. Novelty and technical difficulty: Diff-SNPO seems to be a relatively straightforward combination of CFG and BDPO, which are both existing methods. Given that BDPO addresses the problem of preference learning procedures reducing the likelihood of the preferred examples, it is not very surprising that it would be effective here. Can the authors describe the technical challenges of combining these ideas?
2. It would be great if the preliminary can include a dedicated section on CFG.
3. The paper highlights that Diff-SNPO may suffer from conflicting gradients between positive and negative examples because it uses a single model. Since DPO doesn't have this problem (there are no inverted examples), how does BDPO, which only bounds the loss scale for dispreferred examples, specifically mitigate the problem of interfering gradients?

**Questions:**

See weaknesses.

---

> ### Author Response · Authors · 2025-11-24
> **Response to Reviewer tGQL**
>
> Thank you for your comments. We've just uploaded a new version of the paper in which we incorporated your suggestions. Below we respond to your comments point by point:
>
> **[W-Q1] Novelty and technical difficulty**
>
> **[W-A1]** The novelty and technical contribution of our work lie in two main aspects:
> 1. We identify and analyze the *blurring issue* that emerges when training a naïve version of Diff-SNPO on diffusion models, as illustrated in `Figure 2&3` as well as its underlying cause.
> 2. While *BDPO* has been proposed in the LLM setting to address the reduction in likelihood for preferred samples, it is not directly applicable to diffusion models due to key differences in training formulation. To address this, we derive an *upper bound on the BDPO objective* specifically adapted for diffusion, which is presented in `Appendix A.2`.
>
> **[W-Q2] It would be great if the preliminary can include a dedicated section on CFG**
>
> **[W-A2]** Thank you for the suggestion. We have added a dedicated section on Classifier-Free Guidance (CFG) in `Section 3.2` of the revised manuscript to provide clearer background and context for its role in our method.
>
> **[W-Q3] Clarification on how BDPO mitigates gradient interference in the single-model setting of Diff-SNPO.**
>
> **[W-A3]** Thank you for the thoughtful question. The key idea lies in the asymmetric design of the Diff-SNPO loss. In the conditional branch, the loss on ‘losing’ samples is explicitly down-weighted by the parameter $\lambda$. Without this attenuation, the standard DPO objective in the conditional branch would produce increasingly large gradients to penalize these losing samples as their likelihood decreases, creating a strong rejection force that conflicts with the unconditional branch, whose goal is to increase the likelihood of those same samples. In the unconditional branch, the loss on ‘winning’ samples, which due to label flipping are treated as the ‘losing’ samples for that branch, is similarly scaled down. This prevents the rejection signal in the unconditional branch from overwhelming the conditional branch’s objective of aligning with these winning samples. By bounding these negative contributions in both branches, Diff-SNPO reduces gradient conflict and allows the acceptance gradients from each branch to guide learning without destructive interference.

---

### Author Response · Authors · 2025-11-24
**General Response**

We thank all reviewers for their time and constructive feedback, which has significantly improved our work. Below, we summarize the changes that have been incorporated into the revised version (shown in blue):

- `L48–70` and `Table 1`: In response to Reviewer cffW, we revised the introduction and Table 1 to include a short description of CHATS, better contextualizing our work and contributions within the broader literature on negative-preference alignment algorithms.

- `Section 3.2`: In response to Reviewer tGQL, we added a preliminaries section on classifier-free guidance (CFG).

- `Figure 3` in `Section 4.2`: In response to Reviewer ascV, we added more examples to better illustrate the blurring phenomenon of Naive-SNPO.

- `Table 2` in `Section 5`: In response to Reviewer cffW, we added error bounds for the main quantitative results.

- `Section 5.3`: In response to Reviewer ascV, we added additional ablation results on the effect of the hyperparameters $\lambda$ and $\beta$ on Diff-SNPO.

- `L505-511` and `Table 5`: In response to Reviewer cffW, we added negative preference loss results and additional discussions to better highlight the training–sampling mismatch problem of Diff-NPO.

- `Table 4` in `Section 5.5`: In response to Reviewer f2zF, we added per-iteration runtimes for batch size 512 using $8\times$A6000 GPUs.

- `Table 6` in `Appendix A.3`: In response to Reviewer cffW, we added error bounds for the quantitative results on the HPDv2 dataset.

- `Appendix A.4`: In response to Reviewer ascV, we added additional quantitative results for Diff-SNPO under different ODE solvers.

- `Appendix A.5`: In response to Reviewers f2zF and ascV, we added additional experimental results on the CoProv2 dataset to evaluate the safety alignment of different alignment algorithms.

- `Appendix A.6`: In response to Reviewer ascV, we added the total computational cost for all methods.

---

### Comment · Area_Chair_BsP4 · 2025-11-29

Dear Reviewers,

Authors’ kindly tried to address your concerns. If the responses address your concerns please acknowledge that. If not, please express remaining concerns. Thanks for your efforts!

Best, AC

---

### Meta-Review · Area_Chair_BsP4 · 2026-01-05

**Summary:**

This paper introduces Diff-SNPO (Diffusion Simple Negative Preference Optimization), a unified framework for aligning diffusion models using both positive and negative preference pairs within a single model architecture. The authors identify a "blurring" phenomenon inherent in naive single-model implementations of DPO (due to likelihood reduction of preferred samples) and propose a solution by deriving a tractable upper bound of Bounded DPO (BDPO) adapted for diffusion. By utilizing the conditional and unconditional branches of Classifier-Free Guidance (CFG) to encode positive and negative preferences respectively, Diff-SNPO avoids the computational overhead and model-merging complexities of dual-model approaches like Diff-NPO. Experiments on Pick-a-Pic v2 and HPDv2 demonstrate that the method achieves superior alignment performance and safety scores while reducing training and inference costs by approximately $2\times$.

**Reviewer Concerns:**

The reviewers initially raised concerns regarding novelty, the lack of safety benchmarks, statistical significance, and hyperparameter sensitivity. The authors provided a comprehensive rebuttal that addressed nearly all issues with new experiments and revisions.

Resolved Concerns:Safety and Generality (Reviewers ascV, f2zF): Reviewer ascV (initially rejecting) and f2zF questioned the method's performance beyond aesthetic metrics and the potential safety risks of training negative signals into the unconditional branch. The authors added experiments on the CoProv2 safety benchmark (Appendix A.5). Results demonstrated that Diff-SNPO explicitly trained for safety outperforms Diff-DPO (lower Inappropriate Probability), validating that the unconditional branch training does not compromise safety guidance.

Statistical Significance (Reviewer cffW): In response to requests for rigor, the authors updated quantitative results to include error bars (95% confidence intervals). The revised data confirms that Diff-SNPO consistently outperforms baselines even when accounting for variance.

Ablations and Hyperparameters (Reviewer ascV): The authors added Section 5.3 to include ablations on hyperparameters $\alpha$ and $\beta$, showing robustness. They also added results for four different ODE solvers (DDIM, Euler Discrete, UniPC, DPM Solver) in Appendix A.4, demonstrating consistent performance across samplers.

Computational Cost Clarity (Reviewers f2zF, ascV): The authors corrected the batch size typo (clarifying effective batch size was 512, not 8) and added total GPU hours for all methods in Appendix A.6 to allow direct comparison.

Novelty and Technical Details (Reviewer tGQL): The authors clarified that the novelty lies in the diagnosis of the "blurring" issue in naive SNPO and the derivation of the diffusion-specific BDPO upper bound. They also added the requested preliminaries section on CFG.

**Reviewer Scores:**

Reviewer tGQL: $4 \rightarrow 6$. The reviewer was already "on the fence" and their questions regarding novelty and technical preliminaries were fully addressed.

Reviewer ascV: $2 \rightarrow 6$. This reviewer gave a "Reject" primarily due to a lack of safety benchmarks, limited qualitative examples, and missing ablations. The authors provided all of these items. While the reviewer might remain conservative, the grounds for rejection have been effectively removed.

---

### Decision · Program_Chairs · 2026-01-26

Accept (Poster)